# Accurate ECG Classification Based on Spiking Neural Network and Attentional Mechanism for Real-Time Implementation on Personal Portable Devices

**Yuxuan Xing** [1,2], **Lei Zhang** [1,3], **Zhixian Hou** [1], **Xiaoran Li** [1,3,*], **Yueting Shi** [1,2], **Yiyang Yuan** [4], **Feng Zhang** [4], **Sen Liang** [5], **Zhenzhong Li** [6] and **Liang Yan** [7]

1    School of Integrated Circuits and Electronics, Beijing Institute of Technology, Beijing 100081, China; 3220200655@bit.edu.cn (Y.X.); ZHL666@bit.edu.cn (L.Z.); 3220210820@bit.edu.cn (Z.H.); shiyueting@bit.edu.cn (Y.S.)
2    Yangtze Delta Region Academy of Beijing Institute of Technology, Jiaxing 314019, China
3    Beijing Institute of Technology Chongqing Center for Microelectronics and Microsystems, Chongqing 401332, China
4    Institute of Microelectronics, Chinese Academy of Sciences, Beijing 100029, China; yuanyiyang21@mails.ucas.ac.cn (Y.Y.); zhangfeng_ime@ime.ac.cn (F.Z.)
5    Youth STEM Academy, Beijing 100089, China; liangsen1992@gmail.com
6    CATARC Automotive Test Center (Ningbo) Co., Ltd., Ningbo 315336, China; lizhenzhong@catarc.ac.cn
7    Amileyuan Intelligent Technology (Beijing) Co., Ltd., Beijing 100020, China; yanliang@ciftis.org
*    Correspondence: xiaoran.li@bit.edu.cn

**Abstract:** Electrocardiogram (ECG) heartbeat classification plays a vital role in early diagnosis and effective treatment, which provide opportunities for earlier prevention and intervention. In an effort to continuously monitor and detect abnormalities in patients' ECG signals on portable devices, this paper present a lightweight ECG heartbeat classification method based on a spiking neural network (SNN), a relatively shallow SNN model integrated with a channel-wise attentional module. We further explore the best-optimized architecture, which benefits from leveraging the full advantages of the SNN potential with the attention mechanism to process the classification task at low power and capture prominent features concerning the time, morphology, and multi-channel representations of the ECG signal. Results show that our model achieves overall classification accuracy of 98.26%, sensitivity of 94.75%, and F1 score of 89.09% on the MIT-BIH database, with energy consumption of 346.33 µJ per beat and runtime of 1.37 ms. Moreover, we have conducted multiple experiments to compare against current state-of-the-art methods using their assessment strategies to evaluate our model implementation on FPGA. So far, our work achieves comparable overall performance with all the literature in terms of classification accuracy, energy consumption, and real-time capability.

**Keywords:** electrocardiogram (ECG) classification; spiking neural network (SNN); channel attentional mechanism; portable devices; MIT-BIH database

## 1. Introduction

Cardiovascular disease (CVD) is currently the leading cause of morbidity and mortality worldwide, accounting for approximately 17.3 million deaths annually [1]. It can remain latent in the body for extensive periods and tends to start with occasional asymptomatic arrhythmia, which is characterized by a set of erratic heartbeats. The electrocardiogram (ECG) signal is the most commonly used diagnostic object to examine abnormal heartbeats, as it enables a non-invasive, almost radiation-free, and immediate interpretation of the electrical state generated by the heart beating [2]. However, it is difficult to capture abnormal heartbeats in a short interval, resulting from cardiac arrhythmias with the characteristics of sudden and episodic occurrences. Accordingly, automatic continuous diagnosis of real-time ECG signals is extremely critical to improve the efficiency of CVD care in daily routines.

In recent years, wearable devices for continuous ECG monitoring have sprung up and drawn great interest in the scientific community, as they are easy-to-use, portable, low-cost, and do not require experienced experts [3]. Most works with a lack of further regard to ECG auxiliary diagnosis [4–7] have paid intensive attention to recording the ECG signals or detecting the QRS complex, which contains a Q wave, an R wave, and an S wave, as shown in Figure 1. In addition, some work sent obtained ECG data to remote servers in order to perform a huge amount of calculations that analyze the signal, due to the limited storing and computing abilities of wireless sensor nodes [8,9]. Such solutions introduced new problems associated with transmission speed, data security, and device reliability. To avoid these issues, our approach proposes that the heartbeat classification algorithm is directly implemented on low-power wearable devices for automatic continuous diagnosis in real time.

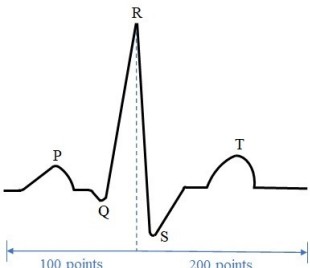

**Figure 1.** The structure of an ECG signal.

Nonetheless, ECG signal recorded by wearable electronic devices is low-frequency and low-amplitude and is inevitably contaminated by a variety of noises and artifacts, such as muscle shaking, electrical interfaces, and even external disturbances caused by poor contact. These noises could cause morphological discrepancies and mask key characteristics, whose reduction is an essential preprocessing step. Presently, available denoising methods for the ECG signal mainly fall into three main categories: conventional digital filtering, adaptive filtering, and morphological filter algorithms dominated by wavelet transform. The effects of these methods on the arrhythmia classification remain unknown [10]. In addition, multiple filter application can reduce the characteristic peak of the ECG signal markedly, and even digital filters for non-stationary signal denoising could cause distorted morphology and loss of significant detailed information [11]. For this issue, we perform wavelet threshold denoising during signal preprocessing. As well, we observe that this method facilitates the subsequent class-imbalanced ECG classification task by means of t-distributed stochastic neighbor embedding (t-SNE) representation.

Arrhythmia classification algorithms in most previous work were typically based on artificial neural network (ANN), recurrent neural network (RNN), or deep convolutional neural network (CNN) with multiple layers and complex recursion structures to improve classification accuracy [12–16]. These approaches result in dramatic increases in hardware costs, deployment complexity, power consumption, and assessment time. Unlike the information processing of CNN, binary event-driven spiking neural network (SNN) neurons are only active when receiving and generating spikes, in contrast to remaining idle at the free event time [17]. The SNN with binary information (0 or 1) transmission consumes energy only when generating spikes, which is well suited for low-power hardware. In addition, SNN processes information with sparse asynchronous spikes, and it makes use of fewer operational quantities than the convolution operation via floats. With respect to this issue, our network model is based on SNN layers with leaky integrate-and-fire (LIF) neurons to achieve higher energy efficiency, as this takes advantage of simple structure, high performance, and reliability.

Paradoxically, prior deployment work based on SNN for arrhythmia classification extracted features without a comprehensive analysis of ECG signal [18–20], which may impede accurate diagnosis. For periodic ECG signals as generalized in Figure 2, there are

substantial variations within each class and great similarities between different classes. As such, extracting heartbeat features requires investigating ECG data thoroughly and systematically. Inspired by [21], the channel attentional module (CAM) cast into the classification task aims to introduce the channel dimension into the feature through automatically learning the weight of each channel, which allows using extracted ECG features with temporal, morphological, and channel characteristics. To alleviate the issue of relatively lower accuracy in previous SNN-based work, we exploit a lightweight CAM to integrate with relatively shallow SNN layers and compensate for the hidden ECG channel information and local pattern extraction in layers. Beyond that, we also consider the embedding location, the type of layer, and hyper-parameter value inside the channel attentional architecture to adopt the best integration strategy, therefore preventing increases in network complexity and power consumption.

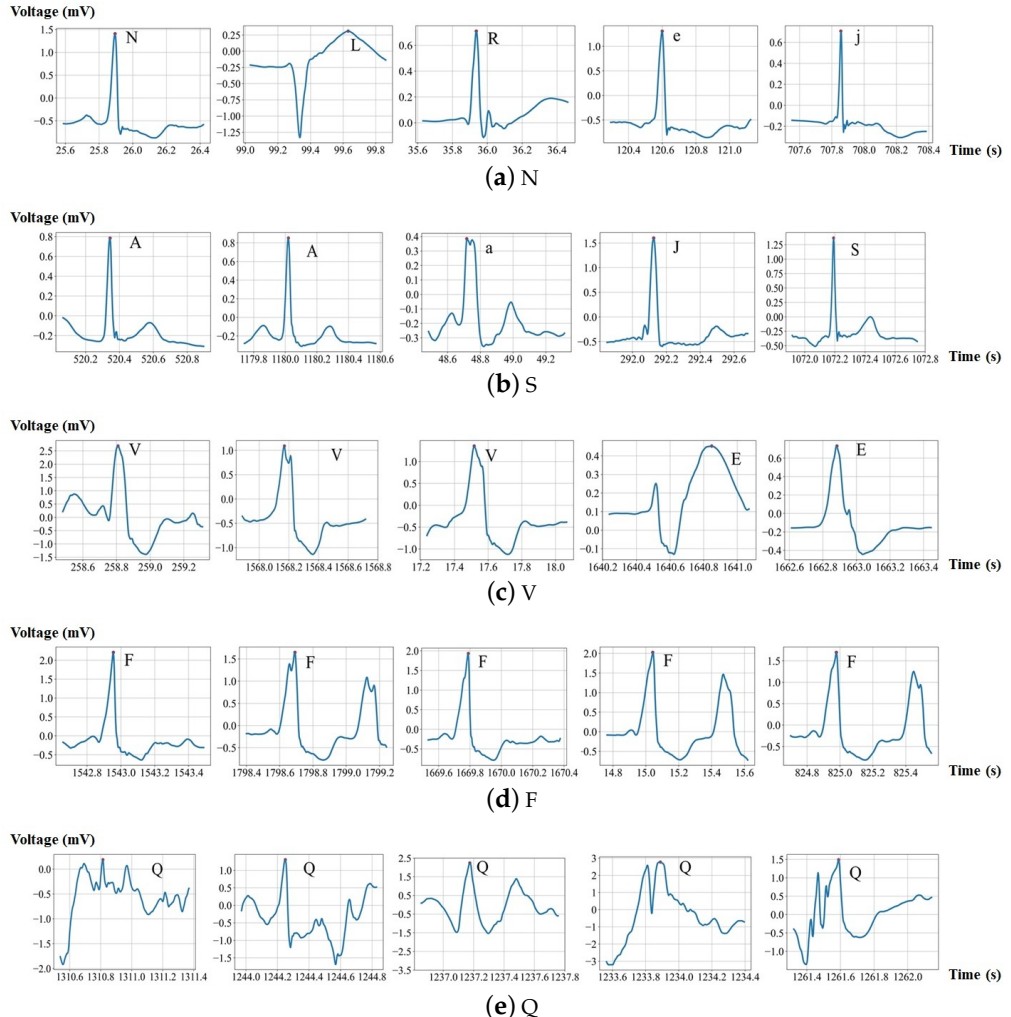

**Figure 2.** Different classes of heartbeats after segmentation. (**a**) Samples in class N. (**b**) Samples in class S. (**c**) Samples in class V. (**d**) Samples in class F. (**e**) Samples in class Q.

The entire workflow of our proposed solution is exhibited in Figure 3. The main contributions of our work are summarized as follows:

1.  We propose a lightweight SNN-based model integrated with an attentional module to incorporate the key channel information into morphological features of time-correlated ECG signals from a global receptive field so as to highlight more discriminative features. In this way, automatic diagnosis of ECG arrhythmia can be achieved at efficient performance with great accuracy and low power consumption.

2. We investigate prevalent noise reduction proposals, among which wavelet threshold denoising is appropriate and effective for ECG signals. Moreover, t-SNE visualization provides great insight to verify the effectiveness of this denoising algorithm for the subsequent automatic classification task, as shown in Figure 4.

3. We conduct extensive experiments under corresponding different assessment strategies to perform ablation studies and select the optimal parameters. Furthermore, our novel algorithm for automatic arrhythmia classification was implemented on FPGA, which accomplishes promising results in contrast with previous state-of-art methods concerning classification accuracy, energy consumption, and real-time capability.

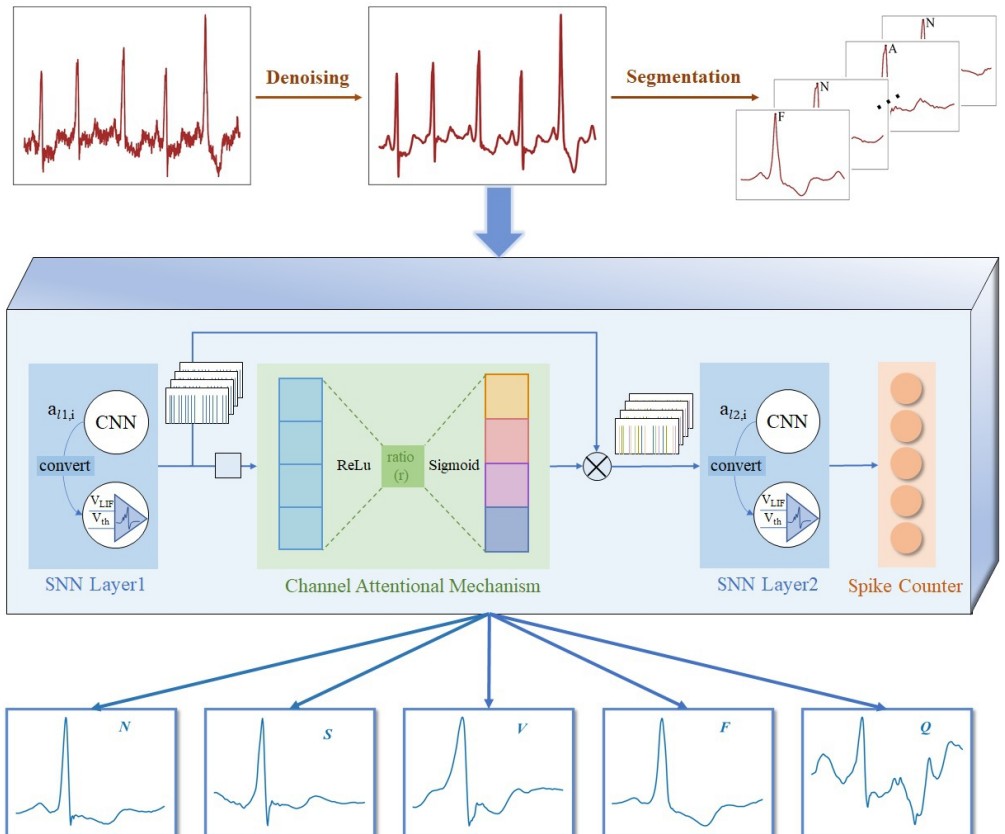

**Figure 3.** System framework of automatic diagnosis for ECG arrhythmia. The last component selects the neuron with maximum spike frequency as the heartbeat's class.

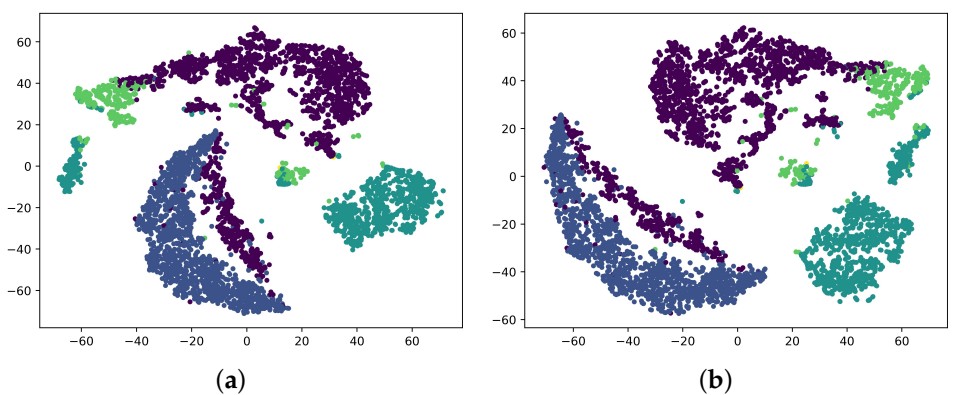

**Figure 4.** The t-SNE visualization for different types of ECG data; different colors represent samples from different types. Denoised ECG data are more easily distinguishable than raw ECG data, especially for V and F types. (**a**) Raw ECG data. (**b**) Denoised ECG data.

## 2. Materials and Related Work

### 2.1. ECG Data

ECG is a low amplitude signal ranging from 10 μV to 5 mV, and its frequency is mainly distributed between 0.5 Hz and 40 Hz. A heartbeat is mainly composed of a P wave, a QRS complex wave, and a T wave, as shown in Figure 1. The P wave reflects atrial contraction, the QRS complex characterized by R-peak represents ventricular depolarization, and the T wave denotes ventricular repolarization gradients.

We employ the Massachusetts Institute of Technology–Beth Israel Hospital (MIT-BIH) arrhythmia database as our clinical ECG signal database. It is one of the most authoritative standard ECG databases in the world, with a total of 48 ECG records annotated by two or more experts. These records contain two-channel ECG signals from 47 different patients with a sampling rate of 360 Hz and a duration of 30 min [22]. According to the Association for the Advancement of Medical Instrumentation (AAMI) recommended practice, four ECG records (102, 104, 107, 217) containing the paced beats are not taken into consideration. For all records, we only use the remaining 44 ECG records by the modified-lead II (MLII). As summarized in Table 1, there are 15 separate classes of heartbeats in the MIT-BIH database, which can be divided into five groups, designated as N (normal), S (supraventricular ectopic beats (SVEBs)), V (ventricular ectopic beats (VEBs)), F (fusion beats), and Q (unknown beats).

**Table 1.** AAMI heartbeat classes and corresponding MIT-BIH heartbeat types.

| AAMI Heartbeat Class | MIT-BIH Heartbeat Annotation | MIT-BIH Heartbeat Types | Number of Heartbeats |
|---|---|---|---|
| | 'N' | Normal beat | 74,546 |
| | 'L' | Left bundle branch block beat | 8075 |
| N | 'R' | Right bundle branch block beat | 7259 |
| | 'e' | Atrial escape beat | 16 |
| | 'j' | Nodal (junctional) escape beat | 229 |
| | 'A' | Atrial premature beat | 2546 |
| S | 'a' | Aberrated atrial premature beat | 150 |
| | 'J' | Nodal (junctional) premature beat | 83 |
| | 'S' | Supraventricular premature or ectopic beat | 2 |
| V | 'V' | Premature ventricular contraction | 6903 |
| | 'E' | Ventricular escape beat | 106 |
| F | 'F' | Fusion of ventricular and normal beat | 803 |
| | '/' | Paced beat | 0 |
| Q | 'f' | Fusion of paced and normal beat | 0 |
| | 'Q' | Unclassifiable beat | 15 |

### 2.2. Signal Denoising

Generally, the dominant noises in ECG signals are power line interference, muscle contractions, and baseline wander, which are in different frequency ranges. Power line interference is a low amplitude signal with an approximate frequency of 50 Hz that hinders the detection of the P wave and the T wave. Muscle contraction is an irregular interference, and its frequency is mainly distributed between 30 Hz and 300 Hz. This kind of noise coincides with ECG signals and causes subtle changes in useful ECG signals to be ignored. Baseline wander is a low-frequency (0.15 Hz up to 0.3 Hz) noise, which affects the PR and ST segments. Therefore, the first step for heartbeat classification is to reduce noises and artifacts in the ECG signal.

For these noise types in ECG signals at different undetermined frequency bands, the most widely used denoising solutions are filter methods and wavelet threshold [23]. Recursive digital filters of the finite impulse response (FIR) [24] has an excellent outcome in attenuation of the known frequency bands, rather than the unknown ones. ECG denoising

requires multiple low-pass and high-pass filters, which could distort the morphology of the ECG signal. Even when using architectures with adaptive filters, this technique has certain constraints and no great advantages for such ECG signals.

With the rapid development of wavelet technology, a range of wavelet transform methods have been applied to obtain pure morphological signal excluding noise [25]. Conforming the frequency distribution of ECG signal and noise at different scales, wavelet transform is applied to the signal, and then the wavelet coefficient of each layer is subjected to thresholding; finally, noise reduction is realized through signal reconstruction. Wavelet transform is a reasonable method when processing non-stationary signals such as ECG signal, as it retains characteristics of the signal and prevents the loss of important physiological details with simple computations. Considering these, we exploit the wavelet threshold method during the denoising process.

### 2.3. Spiking Neural Network

Serving as a low-power alternative to ANN, spiking neural networks (SNN) more closely emulate the biological process with pulse-driven calculation. It is the third generation of neural networks with more biological interpretability that performs numerical simulation and quantitative analysis on biological neural systems [17]. Thus, SNN has been one of the major approaches for neuroscience research, and its applications have become prevalent in medical diagnosis fields [26], such as the medical image analysis, electrogastrogram (EGG) signal recognition [27], electroencephalogram (EEG) signal analysis [28], etc. This is primarily because the spiking neuron model is a valid tool to construct a computational model for detecting and processing signals.

The leaky integrate-and-fire (LIF) model is the most utilized neuron for SNN, and incoming signals have a direct influence on its membrane potential, as generalized in Figure 5. Clearly, the LIF neuron model transmits information effectively by exciting non-differentiable discrete spikes, which is why it cannot train the network by means of standard backpropagation [29].

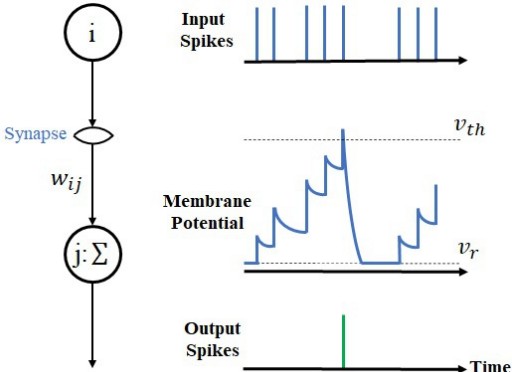

**Figure 5.** The dynamics of an LIF spiking neuron.

Focusing on this problem, Lee et al. [30] proposed a solution that ran a gradient descent algorithm with multiplicative update rules. In addition, Panda et al. [31] built a hierarchical spiking convolutional autoencoder with backpropagation. Similar to the ANN-to-SNN conversion, another popular approach converted the trained CNN model into a spiking architecture, so that SNNs can be exempt from backpropagation during training [26]. These methods used neural membrane potential to approximate differential activation function to run the backpropagation algorithm. Many studies have demonstrated that the converted spiking CNNs have high performance (close to regular CNNs) with fewer operations and less energy consumption, with the aid of which deep CNN is able to be implemented on hardware [26].

## 3. Methology

### 3.1. Wavelet Transform Threshold Denoising

ECG is an unstable weak signal accompanied by a variety of noise and interference in known or unknown frequency bands. Thereby, it is inappropriate to employ the original ECG signal directly. Figure 6 displays the main process of denoising via the wavelet transform method combined with soft thresholding.

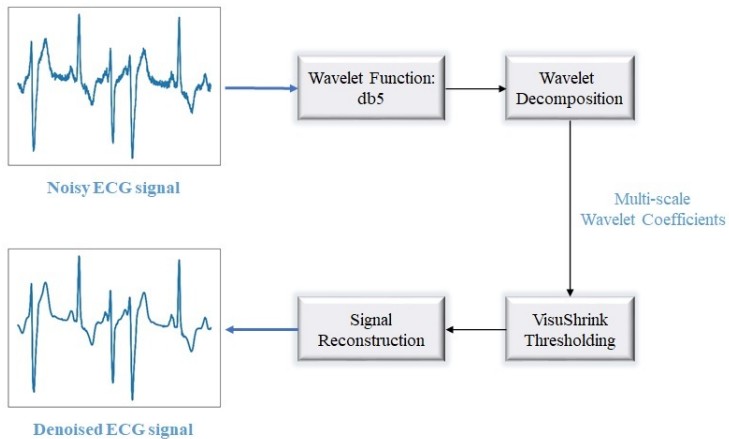

**Figure 6.** The denoising process of an ECG signal.

We selected the Daubechies 5 (db5) wavelet as the wavelet function with a decomposition level of 9 to transform noisy ECG signals into the frequency domain and obtained a set of wavelet decomposition coefficients [23]. If the obtained wavelet coefficient exceeded the soft threshold, it was determined as noise and needed to be discarded. In contrast, if the wavelet coefficient was greater, it could be assumed as the signal and needed to be retained. Based on the soft thresholding of [32], the setting threshold was calculated by the following formula:

$$\lambda = \frac{MAD}{0.6745}\sqrt{2\ln N},\tag{1}$$

where $\lambda$ is the threshold, $MAD$ is the median amplitude of the wavelet coefficients from the first layer, 0.6745 is the adjusted coefficient for standard variance of the Gaussian noise, and $N$ is the length of the signal.

Unlike the hard thresholding, the soft-thresholding function is continuous at the threshold. Thereby, the reconstructed signals are relatively smoothed through the soft thresholding. The amplitude of each wavelet coefficient $|x|$ is compared with the set threshold $\lambda$, as described in the following equation:

$$f(x) = \begin{cases} sgn(x)(|x| - \lambda) & \text{if } |x| > \lambda \\ 0 & \text{if } |x| \le \lambda \end{cases}\tag{2}$$

The return value $sgn(x)$ reflects the plus or minus sign of the parameter $x$, according to the following equation:

$$sgn(x) = \begin{cases} 1 & \text{if } x > 0 \\ -1 & \text{if } x < 0 \end{cases}\tag{3}$$

Finally, through the inverse wavelet transformation, the ECG signal is reconstructed with these processed wavelet coefficients to obtain the denoised signal.

### 3.2. Segmentation ECG Heartbeat

Segmentation of the successive ECG signals is indispensable for extracting morphological features and classifying abnormal heartbeat. In most work, segmenting the ECG signal first requires identifying the R-wave peak [7,25]. At first, the position of the R-wave

peak in each heartbeat can be located by reading the entire annotation provided by the MIT-BIH dataset; it is the strongest peak in each heartbeat. Alongside this, we also adjust the peak values in the annotation file, as several signal points are offset after denoising. Accordingly, there is no need to propose another algorithm for acquiring QRS complexes and detecting the R peak. Second, we remove the first 10 and last 5 consecutive beats to ensure that our analyzed data are from the steady state. Finally, for the sake of extracted feature vectors with the same length and containing P, QRS, and T wave information, based on the determined R-wave peak, we take 100 points forward and 200 points backward as a heartbeat. Figure 2 shows different types of heartbeats after segmentation. Table 1 describes the number of final obtained heartbeats from the 44 ECG records in the MIT-BIH database.

*3.3. Model Design*

3.3.1. SNN-Based Module

For the LIF neurons [29], as detailed in Figure 5, the synapse is a connection between the presynaptic neuron *i* and the postsynaptic neuron *j*. Within the time constant $\tau$, the membrane potential $U_j(t)$ accumulates the leak currents and the input spikes generated by different pre-synaptic neurons *i*. The potential $U_j(t)$ could store the temporal information and is calculated with the following equation:

$$\tau \frac{dU_j(t)}{dt} = -U_j(t) + RI(t),\tag{4}$$

where $\tau$ represents the time constant for the leak current, and $R$ and $I(t)$ represent the input resistance and drive current in the LIF circuit, respectively.

As digital simulation measures continuous current and voltage, differential Equation (4) is converted into a difference equation for computation. From a difference perspective, the membrane voltage is represented as

$$U_j(t) = \lambda U_j(t-1) + \sum_i w_{ij} U_i(t),\tag{5}$$

where $U_j(t)$ is the membrane voltage of postsynaptic neuron *j* at the time *t* and $w_{ij}$ is synaptic weight, which weights the inputs of presynaptic neuron *i*. As the SNN layer with LIF neuron encodes incoming numerical signals into binary spikes, the multiplication of $U_i(t)$ and $w_{ij}$ is turned into addition.

When the membrane voltage $U_j(t)$ exceeds the voltage threshold $v_{th}$, an action potential is fired, and neuron *j* generates a spike. The output spike at time *t* is calculated by:

$$Output(t) = \begin{cases} 1, & \text{if } U_j(t) > v_{th} \\ 0, & \text{otherwise} \end{cases}\tag{6}$$

Immediately afterwards, the membrane potential $U_j(t)$ is dropped to the reset potential $v_r$ and neuron *j* enters the refractory period. To simplify the calculation, we set the reset voltage $v_r$ to 0 without affecting the biological neuron model. During the refractory period, the membrane voltage of postsynaptic neuron *j* stays constant and does not respond to new stimulation until the next cycle.

In the SNN layers, the membrane potential of the LIF neuron approximates the original activation function in convolution, which similarly supports other network layers in CNN, such as the pooling layer and the fully connected layer.

3.3.2. The Channel Attentional Mechanism

After the first SNN layer, the input ECG data are reshaped into the local feature $F = [f^1, f^2, \ldots, f^c] \in R^{1 \times W \times C}$ with a size of $1 \times W$ and a channel number of $C$ (here $W = 140$ and $C = 4$), where $f^s$ denotes the feature vector for the channel *s*. The value of $F$ is the product from all channels, but its channel relationship is ignored and implicit in the

local spatial feature due to the convolution operation. Beyond that, our simple network structure only contains two convolution operations, which cause the receptive field to be relatively small, and the contextual information outside the local region is not available. Inspired by the SE block in [21], we approach the following operations for the limitations of a shallow network, as presented in Figure 7.

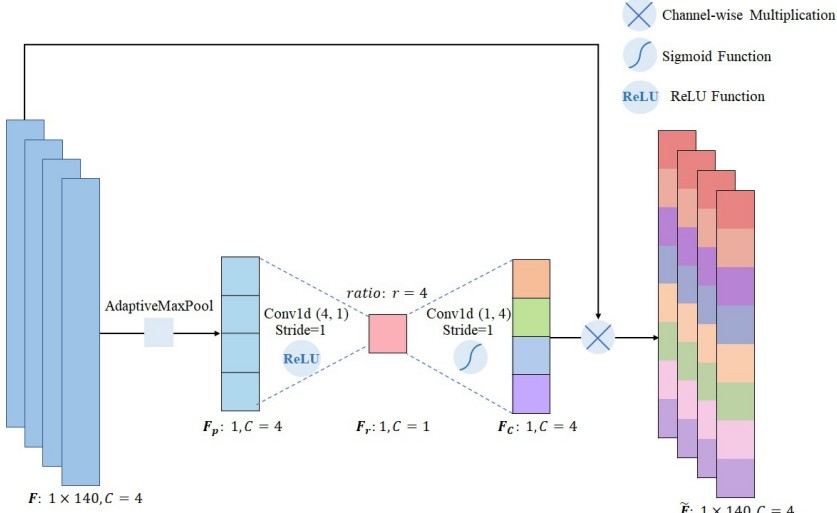

**Figure 7.** The architecture of the channel attentional mechanism, where different colors indicate the importance level of different channels.

First, the vector $F$ enters into the global max-pooling (GMP) layer, so that its output dimension is matched with its own channel number to refine global information and obtain the vector $Fp = \left[ fp^1, fp^2, \ldots, fp^c \right] \left( Fp \in R^{1 \times C} \right)$. In a way, the GMP operation squeezes feature vector $F$ into the one-dimensional feature $Fp$ that represents the importance of each channel within a global perspective, as follows:

$$fp^i = max(f_j^i), \quad j \in [1, N], \tag{7}$$

where $f_j^i$ denotes the $j$th element in the $i$th channel of the feature $F$ and $N$ is the number of the elements in $f^i$.

After that, the feature vector $Fp$ is fed into a bottleneck structure for better utilization of aggregated information produced by the GMP layer. This bottleneck structure is composed of two convolution layers with a kernel size of 1 with a hidden layer between them. With the reduction ratio $r$ (here $r$ is 4), the number of neural units in the hidden layer is first reduced to $\frac{C}{r}$ of the input convolution layer, and then the output convolution layer adopts the same number of neural units as the input convolution layer to obtain a new channel-wise feature $Fc$ with a size of 1 and channel number of $C$. The rectified linear unit (ReLU) layer follows the first convolutional layers, which can introduce sparsity to control overfitting and speed up computing time [33]. After the last convolutional layer, the sigmoid activation layer is utilized, so that $Fc$ contains non-mutually exclusive multiple channel information and is normalized between 0 and 1. These operations are calculated according to the following formula:

$$Fc = \sigma(g(Fp, W)) = \sigma(W_2 \delta(W_1 Fp)), \tag{8}$$

where $\delta$ is the ReLU function, $\sigma$ is the sigmoid function, $W_1 \in R^{\frac{C}{r} \times C}$ and $W_2 \in R^{C \times \frac{C}{r}}$.

Finally, the final feature vector $\tilde{F} = \left[ \tilde{f}^1, \tilde{f}^2, \ldots, \tilde{f}^c \right] \in R^{1 \times W \times C}$ is obtained by the channel-wise multiplication of the feature $Fc$ and the previous feature vector $F$, which

accomplishes adaptive recalibration with the learned channel information to represent the heartbeat, using the following equation:

$$\tilde{F} = Fc \cdot F \tag{9}$$

This CAM is incorporated between two SNN layers, and its integration strategy is discussed in detail in Section 4.2. The specific values corresponding to all aforementioned parameters are shown in Figure 7.

## 4. Experiments and Discussion

### 4.1. Performance Metrics

Our model is trained for 30 epochs under Pytorch 1.7.1, using mean squared error (MSE) loss, a batch size of 32, and the Adam optimizer with a learning rate of 0.001. Following the AAMI recommendations, the evaluation metrics for ECG classification performance in our paper are accuracy (*Acc*), sensitivity (*Sen*), precision (*Pre*), and F1 score (*F*1). These metrics are defined as follows [19]:

$$Acc = \frac{TP + TN}{TP + TN + FP + FN} \tag{10}$$

$$Sen = \frac{TP}{TP + FN} \tag{11}$$

$$Pre = \frac{TP}{TP + FP} \tag{12}$$

$$F1 = \frac{2}{1/Sen + 1/Pre} \tag{13}$$

Here, $TP$ means true positives, $TN$ means true negatives, $FP$ means false positives, and $FN$ means false negatives. $F1$ score comprehensively considers the precision and the recall of the classification model, which is frequently taken as the final evaluation metric.

### 4.2. Model Construction

1. *The t-SNE visualization of denoised ECG data*: The dataset for training and testing consists of 44 ECG recordings: 100, 101, 103, 105, 106, 108, 109, 111, 112, 113, 114, 115, 116, 117, 118, 119, 121, 122, 123, 124, 200, 201, 202, 203, 205, 207, 208, 209, 210, 212, 213, 214, 215, 219, 220, 221, 222, 223, 228, 230, 231, 232, 233, and 234. Five major groups (N, S, V, F, and Q) of heartbeat data are all taken into account, so that our classification method can be extensively implemented in clinical arrhythmia diagnosis. We select the concise F1 score in Equation (13) as the main indicator to assess each class. In addition, to make the most of available ECG data to design the optimal model, all extracted heartbeat data are randomly split into a training set (70%) and a testing set (30%). In Figure 4, we carry out t-SNE visualization to compare the feature separability of the denoised data with the raw data that are selected from records 208 and 232 in the MIT-BIH arrhythmia database.

2. *SNN parameters*: The optimization of the converted SNN is implemented in the SpikingJelly framework under Pytorch, which provides four basic modules for SNN deep learning: Neuron, Layer, Functional, and Encoding. The SNN-based model contains the LIF neuron from the Neuron module and represents each input sample as binary spikes at each timestep (from 1 to T). The timestep T determines the computational amount of each sample with an impact on accuracy and latency. To trade off latency and maximum accuracy by setting the optimal timestep, we explore the effect of the timesteps in LIF neurons on the classification accuracy. As indicated in Table 2, the maximum classification accuracy 98.26% is obtained when the value of the timestep is 7.

**Table 2.** SNN maximum accuracy with respect to the number of timesteps.

| Timesteps | Overall-Acc (%) | N-F1 (%) | S-F1 (%) | V-F1 (%) | Q-F1 (%) |
|---|---|---|---|---|---|
| 1 | 97.45 | 98.69 | 76.14 | 91.73 | 80.56 |
| 2 | 97.49 | 98.71 | 71.16 | 92.97 | 68.82 |
| 3 | 97.74 | 98.82 | 72.46 | 94.44 | 78.13 |
| 4 | 98.09 | 99.06 | 79.33 | 94.74 | 77.07 |
| 5 | 97.42 | 98.66 | 69.55 | 93.02 | 70.74 |
| 6 | 97.71 | 98.83 | 72.45 | 94.17 | 75.32 |
| 7 | **98.26** | **99.13** | **80.67** | **95.40** | **81.16** |
| 8 | 97.59 | 98.74 | 73.42 | 93.45 | 71.29 |
| 9 | 98.06 | 99.01 | 76.86 | 95.42 | 80.85 |
| 10 | 98.17 | 99.08 | 80.68 | 94.87 | 81.34 |
| 11 | 97.85 | 98.88 | 73.25 | 95.04 | 77.97 |
| 12 | 97.83 | 98.90 | 74.15 | 94.56 | 75.89 |
| 13 | 97.94 | 98.97 | 76.10 | 94.39 | 79.02 |
| 14 | 97.56 | 98.72 | 71.88 | 93.59 | 70.43 |
| 15 | 98.16 | 99.07 | 77.24 | 95.59 | 80.85 |
| 16 | 98.19 | 99.08 | 78.99 | 95.50 | 81.82 |

3. *Ablation study for the CAM*: We perform ablation experiments on different configurations of module components to design the optimal structure that integrates our channel attention mechanism into the SNN module.

First, we exploit different squeeze operators in the heartbeat classification tests to investigate the significance level of global average pooling (GAP) and the GMP layer, as well as all of them together. Table 3 suggests that the global max pooling serves as the squeeze operator to perform the best with the fewest parameters for all ECG classes.

**Table 3.** Effect of different squeeze operators in the CAM.

| Squeeze Operator | Overall-Acc (%) | Params | N-F1 (%) | S-F1 (%) | V-F1 (%) | F-F1 (%) |
|---|---|---|---|---|---|---|
| GAP | 98.08 | 435,336 | 99.01 | 79.80 | 94.71 | 80.00 |
| GMP | **98.26** | **435,336** | **99.13** | **80.67** | **95.40** | **81.16** |
| GAP + GMP | 98.19 | 435,344 | 99.10 | 79.77 | 95.31 | 78.93 |

Next, we explore the location choice of the channel attention mechanism integrated into the existing SNN-based architecture. The integrated CAM is generic to build blocks, but it is not necessarily appropriate for all of them. To adequately model channel-wise dependencies in features, the CAM requires a relatively high number of channels to be processed. Apparently, it is undesirable to place the CAM before SNN Layer1. We consider three variant designs, as shown in Figure 8: (1) CAM-Between, in which the attentional module is placed between SNN Layer1 and SNN Layer2; (2) CAM-After, in which the attentional module is placed after SNN Layer2, and (3) CAM-All, in which the attentional module is placed both between the two SNN layers and after SNN Layer2. Relative to the CAM-After block in which the CAM is followed by the fully connected classifier, the CAM-Between block excites more features with channel information. In the CAM-All block, the effects produced by the CAM between the two SNN layers and after SNN Layer2 are complementary.

It currently remains challenging to report a rigorous theoretical analysis for multi-channel representations learned by deep neural networks [21]. Thence, we conduct a series of ablation experiments to examine the practical function of the integrated CAM position, and the experimental results are listed in Table 4. The CAM-Between block performs the best with the least increase of model parameters, whereas the CAM-After integration leads to dramatically worse performance and even drops

below the baseline module (containing only two SNN layers). The CAM-All block is not quite as accurate as the CAM-Between block due to the effect of the CAM after SNN Layer2. This demonstrates that a careful integration strategy is quite significant for the channel attention module to bring more performance benefits.

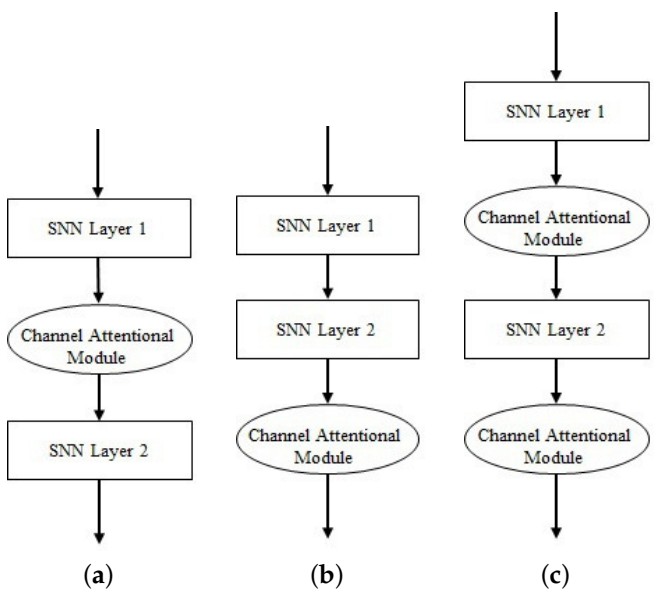

**Figure 8.** Three integration designs for the CAM. (**a**) CAM-Between block. (**b**) CAM-After block. (**c**) CAM-All block.

**Table 4.** Effect of the CAM with different integration strategies.

| Integration Design | Overall-Acc (%) | Params | N-F1 (%) | S-F1 (%) | V-F1 (%) | F-F1 (%) |
|---|---|---|---|---|---|---|
| Baseline (SNN) | 98.09 | 435,328 | 99.05 | 78.35 | 94.99 | 77.00 |
| CAM-Between | **98.26** | 435,336 | **99.13** | **80.67** | **95.40** | **81.16** |
| CAM-After | 97.70 | 435,360 | 98.84 | 71.15 | 94.10 | 76.47 |
| CAM-All | 98.16 | 435,368 | 99.06 | 80.05 | 95.39 | 80.47 |

For the final step, based on the current construction with the known integrated location and squeeze operator, we conduct experiments on a range of different values for hyperparameter $r$, which is the channel reduction ratio defined in Equation (8). The comparison in Table 5 shows that tuning the value of $r$ to 4 performs better with the fewest model parameters, although setting it to 2 is effective.

**Table 5.** Effect at different reduction ratios.

| Ratio $r$ | Overall-Acc (%) | Params | N-F1 (%) | S-F1 (%) | V-F1 (%) | F-F1 (%) |
|---|---|---|---|---|---|---|
| 2 | 98.22 | 435,344 | 99.10 | **80.97** | 95.22 | 79.90 |
| 4 | **98.26** | **435,336** | **99.13** | 80.67 | **95.40** | **81.16** |

In Figure 9, we observe that the integrated CAM could improve the feature extraction ability and achieve a better outcome in contrast to the original. In particular, the CAM-Between block has the optimal integration strategy to prevent increases in model complexity and the highest classification accuracy, and it takes almost negligible computational cost to maximize efficacy. Additionally, the F1 score for each type of heartbeat is indeed greatly improved, as displayed in Table 4. However, similar to most other work [13,18,34], class Q is poorly classified. This is due to many variations within class Q and few training samples of class Q, as generalized in Figure 2 and Table 1.

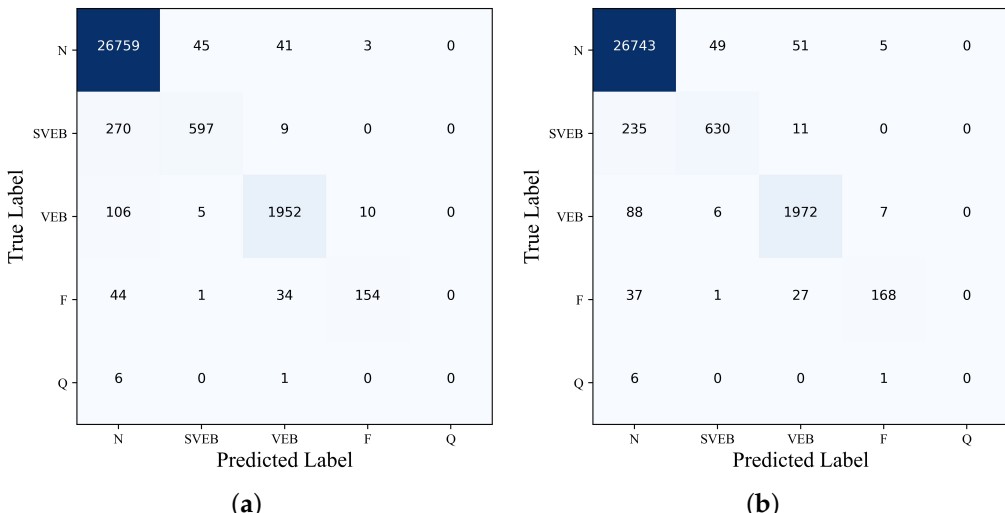

**Figure 9.** The confusion matrix heatmaps of heartbeat classification: (**a**) confusion matrix for the original SNN-based model; (**b**) confusion matrix for the SNN-based model integrated with the CAM.

*4.3. Comparison with Previous Work*

For the sake of a comprehensive and fair evaluation in specific details when compared with seven state-of-art methods, we adopt their corresponding evaluative strategies. Against these previous methods, our work performs comparably under most strategies, as indicated in Table 6. These different evaluative strategies include the "70:30" strategy, Inter-patient strategy, and Patient-specific strategy. Particularly, the work marked with an asterisk is assessed for seven types of heartbeats, so we do not adopt its corresponding strategy. In addition, for a detailed view of the results on the hardware, we implement the classification model on Artix-7, which provides high performance in power consumption and packaging miniaturization.

The "70:30" strategy intends that all heartbeats from 44 ECG records are randomly divided into a training set (70%) and a testing set (30%). Table 6 shows that our overall performance for identifying five types of heartbeats is nearly close to [15] with an 11-layer deep CNN structure.

Referring to the Inter-patient strategy to make similar proportions for each type of heartbeats, the training dataset contains 22 ECG records (101, 106, 108, 109, 112, 114, 115, 116, 118, 119, 122, 124, 201, 203, 205, 207, 208, 209, 215, 220, 223, and 230) and the testing dataset contains the remaining ECG records (100, 103, 105, 111, 113, 117, 121, 123, 200, 202, 210, 212, 213, 214, 219, 221, 222, 228, 231, 232, 233, and 234). Relative to [19], despite our work producing a slightly higher power consumption, there are higher scores in terms of the overall accuracy and F1 scores for class N and class V.

Conforming to the Patient-specific strategy, the ECG dataset is partitioned into two groups, and they are referred to as the DS1 group and the DS2 group. The DS1 group consists of the first 20 ECG records (100, 101, 103, 105, 106, 108, 109, 111, 112, 113, 114, 115, 116, 117, 118, 119, 121, 122, 123, and 124), which include typical ECG abnormalities in routine clinical practice. The DS1 group consists of the remaining 24 records (200, 201, 202, 203, 205, 207, 208, 209, 210, 212, 213, 214, 215, 219, 220, 221, 222, 223, 228, 230, 231, 232, 233, and 234), which include complex arrhythmias, such as ventricular arrhythmias, junctional arrhythmias, and conduction abnormalities. Under the Patient-specific strategy, all ECG signals in DS1 plus the first five minutes of ECG signals in DS2 are utilized as the training dataset to train a model for the patient's own ECG data. The rest of ECG data serve as the test data. In particular, [13,14,20] are software-based solutions, and [18] provided their implementation on ARM Cortex A53.

**Table 6.** Performance comparison of our proposed approach with other previous work. The values in parentheses denote comparisons with our approach. And the work marked with an asterisk classified seven types of heartbeats.

| Approach | Evaluative Strategy | Acc (%) | Sen (%) | Pre (%) | F1 (%) | Energy per Beat | Time per Beat (s) |
|---|---|---|---|---|---|---|---|
| Pandey et al. [15] (11-layer deep CNN) | "70:30" (N, S, V, F, and Q class) | 98.30 (+0.04) | Average: 95.51 (+0.76) | Average: 86.06 (+1.37) | Average: 89.87 (+0.78) | - | - |
| Yan et al. [19] (SNN) | Inter-patient (N, S, V, and F class) | Average: 90 (−2.07) | N: 92 (−6.37) V: 77 (+7.96) | N: 97 (+3.77) V: 59 (−17.46) | N: 94.43 (−1.3) V: 66.81 (−5.75) | Power: 181 mW | - |
| Hu et al. [12] (MLP) | Patient-specific (V class) | 94.8 (−3.4) | 78.9 (−9.4) | 75.8 (−17.86) | 77.3 (−13.6) | - | - |
| Kiranyaz et al. [14] (FFT, CNN) | Patient-specific (V class) | 98.6 (+0.4) | 95 (+6.7) | 89.5 (−4.16) | 92.2 (+1.3) | 37 mJ [18] | 41 ms [18] |
| Ince et al. [13] (Wavelet, MLP) | Patient-specific (V class) | 97.6 (−0.6) | 83.4 (−4.9) | 87.4 (−6.26) | 85.4 (-5.5) | 4 mJ [18] | 4.5 ms [18] |
| Kolagasioglu et al. [20] * (Wavelet, SNN) | Cluster seven beat types | 95.5 (−2.7) | - | - | - | 614 μJ [18] | 994 ms [18] |
| Amirshahi at al. [18] (SNN) | Patient-specific (V class) | 97.9 (−0.3) | 80.2 (−8.1) | 97.3 (+3.64) | 88 (−2.9) | 1.78 μJ | 200 ms |
| **Proposed** (SNN+CAM) | "70:30" (N, S, V, F, and Q class) | Overall-Acc: 98.26 | Average: 94.75 | Average: 84.69 | Average: 89.09 | 346.33 μJ | **1.37 ms** |
| | Inter-patient (N, S, V, and F class) | Overall-Acc: 92.07 | N: 98.37 V: 69.04 | N: 93.23 V: 76.46 | N: 95.73 V: 72.56 | Power: 246 mW | **1.32 ms** |
| | Patient-specific (V class) | 98.2 | 88.3 | 93.66 | 90.9 | 324.51 μJ | **1.32 ms** |

More specifically, our classification accuracy for class V is 3.4% higher than [12] and 0.6% higher than [13], and its F1 score is 13.6% higher than [12] and 5.5% higher than [13]. Although just below 0.4% for the accuracy of class V, our work is superior to [14] in terms of energy consumption and precision, in which the energy consumption is lower by two orders of magnitude, and the precision of class V is 4.16% higher. Compared to the similar SNN-based solution [20], there is better performance in classification accuracy and energy consumption. In contrast to [18] built by the simulations circuit, despite no good result in energy consumption, our work performs well, with higher accuracy and F1 score. Regarding time consumption, our classification for each heartbeat requires the shortest time among all work. In other words, our proposed arrhythmia diagnosis architecture takes the shortest time to accomplish high accuracy relative to other methods with multiple convolutional network layers and low power consumption compared with other SNN-based work.

As indicated in Figure 10, we can conclude that our proposed arrhythmia diagnosis approach has overall comparable performance in terms of classification accuracy, energy consumption, and real-time capability. This primarily benefits from the SNN are the potential to process the classification task at low power, the wavelet threshold denoising to strengthen feature separability and promote subsequent ECG classification, and the integration design for the CAM to maximize efficacy with almost negligible computational cost.

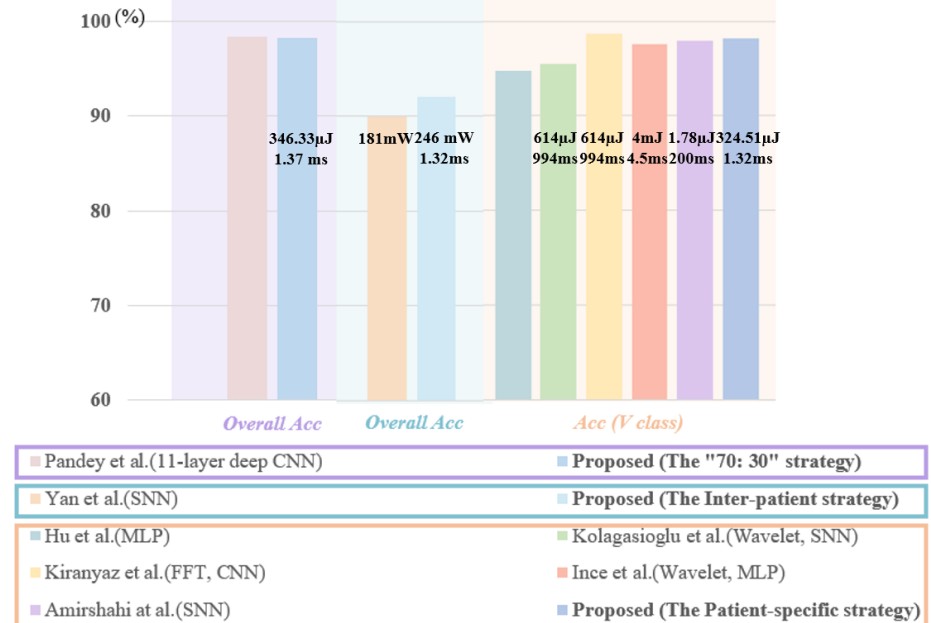

**Figure 10.** The performance comparison with other methods adopting the "70:30" strategy, Inter-patient strategy, and Patient-specific strategy.

## 5. Conclusions

In this paper, we proposed a suitable and effective heartbeat classification algorithm for implementation on low-power hardware composed of the CNN-to-SNN conversion ensemble with the channel attentional mechanism. Furthermore, we performed a range of ablation studies to search for the optimal solution for tuning hyper-parameters and configuration within the SNN-based module and the CAM. On the whole, a wide range of experiments illustrated the effectiveness and feasibility of our classification algorithm, which achieves comparable performance versus previous state-of-the-art work under different evaluative strategies. As a result, our model is appropriate to be implemented on low power and portable devices to diagnose ECG abnormalities automatically in real time.

**Author Contributions:** Conceptualization, Y.X. and Y.S.; data curation, Y.X., Z.H. and Y.Y.; formal analysis, Y.X. and F.Z.; funding acquisition, Z.L. and L.Y.; investigation, Y.X. and L.Z.; methodology, Y.X.; validation, Y.X., L.Z. and Z.H.; visualization, X.L. and S.L.; writing—original draft, Y.X.; writing—review and editing, X.L., Y.S. and S.L. All authors have read and agreed to the published version of the manuscript.

**Funding:** This work was supported by the National Natural Science Foundation of China under Grant 62101038 and Grant 61801027.

**Data Availability Statement:** Not applicable.

**Conflicts of Interest:** The authors declare no conflict of interest.

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
