# Peer review of "Accurate ECG Classification Based on Spiking Neural Network and Attentional Mechanism for Real-Time Implementation on Personal Portable Devices"

_electronics, doi:10.3390/electronics11121889_

Round 1

Reviewer 1 Report

Recognition of heart pathological states by electrocardiogram on wearable devices is of great importance for modern healthcare. In this problem, t is necessary to ensure not only the accuracy of classification, but also the energy efficiency of the algorithms. The authors solve the tasks using a third-generation neural network (SNN).

This work is done at a high scientific level. The abstract gives a brief and clear description of the results. The main text is well written and illustrated. A comparison of the achieved results with modern publications is given, authors show that their results are of top level. I believe that the article can be published as is.

Author Response

Thanks a lot for your kind letter. We deeply appreciate your review and recognition of our research work. Once again, thank you for your hard work and consideration.

Reviewer 2 Report

Dear colleagues,

this paper presents a a lightweight ECG heartbeat classification method based on Spiking Neural Network. The authors show that they have achieved overall classification accuracy of 98.26%, sensitivity 9 of 94.75%, and F1 score of 89.09% on the MIT-BIH database, with energy consumption of 346.33µJ per 10 beat and runtime of 1.37ms. This performance is comparable with current state-of-the-Art in literaure.

The paper is well organized. I think it can be published in this form.

Best regards

Ha Duong Ngo

Author Response

Thank you very much for reviewing this manuscript in your busy schedule. We are very pleased and honored by your recognition of our research work. Thank you very much for your consideration and hard work.

Reviewer 3 Report

This paper present a lightweight ECG heartbeat classification method based on Spiking Neural Network (SNN), a relatively shallow SNN model integrated with the channel-wise attentional module. They further explore the best-optimized architecture, which benefits from leveraging the full advantages of the SNN potential with the attention mechanism to process the classification task at low power and capture prominent features concerning the time, morphology, and multi-channel representations of ECG signal. The developed technique has been well stated in the manuscript, with proper simulation tests. There are some comments regarding the details of the manuscript:

1.      Please remove the grammatical mistakes from the whole paper, even in the abstract.

2.      Please synchronize the abbreviations in the manuscript, i.e., a few terms are not abbreviated as well.

3.      Please provide the references for the equations that do not belong to you.

4.      Please discuss the complexity of the proposed algorithms and the benchmark algorithm.

5.      It is appreciated if the authors can quantify the advancement of the developed method, compared to the conventional ones in a general case.

6.      In the abstract authors have claimed that “we have conducted multiple experiments against the current state-of-the-art methods using their assessment strategies to evaluate our model implementation on FPGA”; however, in the results, I did not find any graphical representation of the comparative analysis with some recent benchmark scheme (Instead of Table 6). It is better to add them as well.

7.      Overall, the quality of the paper is good, but revisions are required to improve the quality of the manuscript up to the mark.

Reviewer 4 Report

This paper proposes a lightweight SNN-based model integrated with an attention module to incorporate key channel information into morphological features of time-correlated ECG signals from the global receptive field and highlight more discriminative features to efficiently achieve automatic diagnosis of ECG arrhythmia with high accuracy and low power consumption We show that it is possible to achieve this goal.

This achievement is recognized as a significant contribution to improving battery sustainability and reliability in ECG arrhythmia diagnosis by wearable devices.

In the discussion of the implementation results, I would like to request in-depth discussion of the following two points

1) The architecture with CAM placed between SSN Layer1 and SSNLayer2 had the highest accuracy. It is necessary to explain what the reason is in terms of the role of the CAM.

2) In the comparison with 7 previous studies, it is necessary not only to compare the accuracy and energy consumption, but also to discuss what the reason is contributed by the proposed architecture.
